# Effect of Fasting and Refeeding on Juvenile Leopard Mandarin Fish *Siniperca scherzeri*

**DOI:** 10.3390/ani12070889

**Published:** 2022-03-31

**Authors:** Yi-Oh Kim, Sung-Yong Oh, Taewon Kim

**Affiliations:** 1Chungcheongbuk-do Inland Fisheries Research Institute, Chungju 27432, Korea; kimio@korea.kr; 2Marine Bio-Resources Research Unit, Korea Institute of Ocean Science & Technology, Busan 49111, Korea; syoh@kiost.ac.kr; 3Program in Biomedical Science and Engineering, Inha University, 100 Inha-ro, Michuhol-gu, Incheon 22212, Korea; 4Department of Ocean Sciences, Inha University, 100 Inha-ro, Michuhol-gu, Incheon 22212, Korea

**Keywords:** aquaculture, body composition, blood chemistry, starvation, compensatory growth

## Abstract

**Simple Summary:**

The development of live prey feed domestication techniques for the carnivorous leopard mandarin fish (*Siniperca scherzeri*) requires establishing nutritionally complete diets and research into optimal feeding regimes. Accordingly, the purpose of this experiment was to investigate the effects of varied feeding regimes for *S. schezeri* on compensatory growth. Individuals were divided into a daily feeding control group (C), a 5-day fasting group (F5), a 10-day fasting group (F10), and a 14-day fasting group (F14). The results demonstrated that after four weeks of treatment, all experimental groups experienced full compensatory growth and reached the control group weight. Therefore, this information will be valuable for effective management when fasting is necessary in *S. schezeri* aquaculture.

**Abstract:**

To verify the effect of fasting on juvenile leopard mandarin fish (*Siniperca scherzeri* mean weight, 14.7 g), compensatory growth, body composition, and blood content of juveniles were investigated for six weeks following two-week feeding treatments: fed continuously (control), and fasted for 5 (F5), 10 (F10) and 14 days (F14). Full compensatory growth was evident after four weeks of food resupply in all fasting groups. Specific growth rate, feeding rate, and feed efficiency in all fasting groups were significantly higher than those of the control after the first 2 weeks of food resupply. At the end of fasting, the lipid content, ratio of lipid to lean body mass, hepatosomatic and viscerosomatic indices in all fasting groups, or total cholesterol content in F14 significantly decreased compared to the control. These results indicated that juvenile leopard mandarin fish subjected to 5–14 days of food deprivation could achieve full compensatory growth after feeding resumption for 4 weeks and that the morphological and biochemical indices, as well as body and blood composition, remained comparable to the control group after the completion of the study under our experimental conditions.

## 1. Introduction

Fish farmers employ fasting or limited feeding strategies in short- and long-term durations to reduce mortality or stress during water quality degradation, disease outbreaks, sorting, transferring, or harvesting or to simply improve feeding efficiency and reduce costs [1,2,3,4,5,6]. In contrast to fish fed continuously, those subjected to restricted feeding or fasting can experience greater growth once satiation feeding is resumed [3,5,7,8]. Such tendencies to return to original weights or growth trajectories are referred to as compensatory or catch-up growth [2,6,7,8,9,10,11].

Depending on the target species, compensatory growth efficiency can vary with type, duration, and intensity of fasting performed prior to food resupply [5,11,12]. Furthermore, compensatory growth can increase feeding efficiency and growth rates while lowering feed and labor costs, ultimately translating environmental improvements and economic gains for farmers. Additionally, numerous studies have aimed to standardize rearing patterns through the application of limited food supply methods for reducing stress due to various unpredictable yet frequent events (e.g., environmental degradation, high/low water temperatures, disease, handling, etc.), thereby developing appropriate feeding regimes for realistic aquaculture settings [5,6,13,14,15,16,17].

The leopard mandarin fish, *Siniperca scherzeri* Steidachner, 1832, is an important freshwater fish species with a high commercial value around East Asia, including China, Korea, and Vietnam [18,19,20]. It is harvested from fish farms in commercial sizes ranging from relatively small 100 g to over 300 g, depending on consumer preferences in Korea. Its high potential as a farmed freshwater fish derives from its excellent taste, high market price, rapid growth, and high resistance to disease [21,22]. It has been established that fish growth and feeding efficiency depend on food quantity and quality [23,24], which, in turn, are dependent on the rearing environment, food composition, form, and feeding method [25,26]. A number of practical feeds have recently been developed in connection with nutritional studies of the *S. scherzeri* for commercial aquaculture [9,20,27,28,29,30]. Addition research has also been conducted on feeding regimes, such as frequency [31], feeding ratio for satiation level [32], and feed rate for fish weight [26]; however, to date, no research has been conducted on compensatory growth.

Accordingly, the purpose of the present study was to determine the effects of varying durations of post-fasting satiation feeding on the feed utilization, body composition, and blood composition of juvenile *S. scherzeri*, as well as the strength of compensatory growth. The ultimate goal was to develop an effective and economically optimal feeding supply strategy in *S. scherzeri* farms.

## 2. Materials and Methods

### 2.1. Fish and Rearing Condition

The experiment was conducted using juvenile *S. scherzeri* raised at the Inland Fisheries Research Institute in Chungcheongbuk-do, Chungju, South Korea. Fish were reared in an indoor recirculation system consisting of a submerged nitrifying filter (3000 L capacity), foam separator (100 L), and 20 fiberglass circular tanks (300 L) and were acclimated for 2 weeks prior to initiating the experiment. During the acclimation period, fish were fed the same experimental dry pellets (55.4% crude protein, 14.1% crude lipid), which were used in the previous study [26], to apparent satiation twice daily (09:00 and 17:00). The fiberglass circular tanks were aerated to provide sufficient dissolved oxygen, whereas water temperature, ammonia, nitrous acid, and pH were monitored daily. Experimental temperatures ranged from 26.4 to 27.5 °C, dissolved oxygen concentrations were ≥7.6 mg L^−1^, ammonia concentration was ≤0.35 mg L^−1^, nitrous acid concentration was ≤0.18 mg L^−1^, and pH ranged from 6.7 to 7.3 comparable with previous studies [26,31,32].

### 2.2. Experimental Design and Management

Following acclimatization, 300 juveniles (initial mean body weight: 14.7 g) were randomly assigned to one of four different feeding groups. Each feeding group consisted of 5 tanks (i.e., replicates), and 15 juveniles were stocked per tank. Control group (C) was fed to apparent satiation twice daily (09:00 and 17:00) during the experimental periods, whereas the three remaining groups fasted for: 5 days (F5) from days 10–14 after initiation of the experiment; 10 days (F10) from days 5–14; 14 days (F14) from days 1–14 before satiation refeeding in week 3. The three fasting groups were fed to satiation same as control. Daily feeding was performed by dropping pellet (the same as during the acclimation period) into each tank every few minutes until the fish ceased to eat due to satiation. Care was taken to ensure that all pellets provided were consumed by the fish. The container of the pellets was weighed before the first feeding and after the last feeding daily. Uneaten pellets were collected from each tank by siphon and were dried in the oven to a constant weight. The actual daily consumed feed amount was determined as the difference in weight between the feed supplied into the tank and the uneaten feed removed by siphon. The experiment lasted 8 weeks.

Prior to the experiment, all the fish in each group were fasted for 24 h for gut evacuation, followed by anesthesia with a 2-phenoxyethanol solution (150 mg L^−1^; Sigma, St. Louis, MO, USA) to minimize the stress before body weight and body length measurement. Before measurement, all excess water on the fish skins was removed by blotting with a paper towel and then measured to the nearest 0.01 g and 0.1 cm.

To evaluate the effect of fasting and refeeding, all fish in each group were bulk-weighed at the end of weeks 2, 4, and 6 and at the end of the experiment (i.e., week 8), individual body weight and body length were measured using the same method above. Fish were not fed on the day of measurement.

### 2.3. Analysis of Blood and Body Content

The blood composition, hepatosomatic index (HSI), viscerasomatic index (VSI), protein, lipid, ash, and moisture content of the four feeding groups, as well as the lipid:lean body mass ratio (lipid/LBM) were measured at the ends of weeks 2 and 8. At the end of week 2, 2 tanks in each feeding group were randomly selected, and 7 fish per tank were randomly subjected to blood composition, HSI, and VSI analyses, while the remaining 8 fish per tank were subjected to whole-body analysis. At the end of week 8, the fish in the remaining three tanks of each feeding group were sampled for analysis of blood and body compositions using the same method as at end of week 2. Fish were anesthetized in 150 mg/L of 2-phenoxyethanol solution for 1 min after the measurement of body weight or body length. Then, they were stored at −40 °C for the analysis of body composition.

Before sacrificing the fish, blood samples of the fish were collected from the tail artery using a heparinized syringe following anesthesia with 2-phenoxyethanol solution (150 mg L^−1^). The blood collected was centrifuged at 8870× *g* for five minutes to extract serum, after which hemoglobin (Hb), glucose (GLU), total cholesterol (TCHO), glutamic oxaloacetic transaminase (GOT), glutamic pyruvic transaminase (GPT), and high-density lipoprotein cholesterol (HDLC) in the extracted serum were measured with a DRI-CHEM NX500i (Fujifilm Co., Tokyo, Japan). Immediately after blood sampling, the sacrificed fish were also dissected, and the weights of the liver and viscera were measured to determine HIS and VSI.

Prior to whole-body analysis, fish samples were freeze-dried and homogenized. Proximate analysis was conducted according to the AOAC methods [33]. The crude protein content was determined with the Kjeldahl method using the Auto Kjeldahl System (Buchi, Flawil, Switzerland). After 24 h of desiccation in a dry oven at 105 °C, the moisture content was measured. Crude fat was measured using the ether-extraction method, and crude ash content was determined following 4 h of combustion at 600 °C.

### 2.4. Growth Parameters and Statistical Analysis

The following growth parameters were calculated: specific growth rate (SGR, % d^−1^) = 100 × (lnW_f_ − lnW_i_)/t, feeding rate (FR, % body weight d^−1^ = 100 × C/[(W_f_ + W_i_)/2]/t, and feed efficiency (FE, %) = 100 = (W_f_ − W_i_)/C where W_f_ and W_i_ are final and initial weights (g), t is the feeding duration (day) and C is total feed consumption (g) during t days.

All statistical analyses were performed in SPSS *v.*20 statistical software (SPSS Michigan Avenue, Chicago, IL, USA). Each replicate was considered an experimental unit, and therefore, the mean value obtained from a replicate within each feeding group was used as a data unit [3,5,8,34]. At the beginning and week 2, all feeding groups had 5 replicates (*n* = 5), while 3 replicates (*n* = 3) were used after week 2. One-way analysis of variance (ANOVA) was separately used to compare initial body weight, blood, and body contents among the four feeding groups, and when significant differences were identified, Tukey’s multiple range test (*p* < 0.05) was used to verify the significance between the means. Levene’s test was used to verify the homogeneity of variance prior to ANOVA, and percent data were arcsine-transformed. Repeated measures analysis of covariance (ANCOVA) with one factor was used to determine the effects of four feeding groups on body weight from week 2 to week 8 and on SGR, FR and FE from week 3 to week 8. In the ANCOVA of body weight, initial body weight was used as a covariate to remove any biases resulting from the allometric relationship between body weight and growth [3,5,8,34]. In the ANCOVA of SGR, FR, and FE, body weights at the beginning of weeks 3, 5, and 7 were used as covariates to assess growth potential, as it returns to the original growth trajectory [3,5,8,34].

## 3. Results

During the experimental period, no mortality was observed in any of the experimental groups. Figure 1 shows the body weight change of *S. scherzeri* throughout the experiment. At the onset of the experiment, no significant difference in body weight was detected among all experimental groups (*p* > 0.05), although the body weights of F5, F10, and F14 immediately following the experimental fasting periods were all significantly lower than those of the control group (F = 53.919, df = 3, *p* < 0.0001). Similarly, fish weights were significantly lower among F5, F10, and F14 following the first two weeks of feed resupply (week 4) compared with the control group (F = 14.358, df = 3, *p* < 0.001). Four weeks (week 6) and six weeks (week 8) after feed resupply, the weight of the fish did not differ significantly from that of the control group (week 6, F = 2.828, df = 3, *p* = 0.107; week 8, F = 1.75, df = 3, *p* = 0.234).

SGRs (F = 64.479, df = 3, *p* < 0.001) of all the fasting groups during the first two weeks of feed resupply (weeks 3–4) were significantly greater than the control (Figure 2). During the second two weeks of feed resupply (weeks 5–6), F14 showed a significantly higher SGR than the control group (F = 9.655, df = 3, *p* < 0.01), whereas no such difference was observed for F5 and F10. SGRs during the final two weeks (weeks 7–8) were not significantly different between the experimental groups (F = 1.148.3, df = 3, *p* = 0.3871); however, the overall SGR over the six weeks of feed resupply (weeks 3–8) was significantly higher in the fasting groups (F = 44.21, df = 3, *p* < 0.001), with F14 exhibiting the highest SGR.

During the first two weeks of feed resupply (weeks 3–4), FRs in all fasting groups were significantly higher than in the control group (F = 33.865, df = 3, *p* < 0.001; Figure 3). During the second two weeks of feed resupply (weeks 5–6), F14 showed significantly higher FR than the control group, while F5 and F10 did not differ from the control group (F = 7.707, df = 3, *p* = 0.01). The FR during the final two weeks of resupply (weeks 7 to 8) was not significantly different between the experimental groups (F = 1.034, df = 3, *p* = 0.428). The overall FR during the six weeks following the fasting period (weeks 3–8) showed the highest FR in F14, followed by F10, F5, and the control group (F = 20.103, df = 3, *p* < 0.001).

The FEs in all fasting groups during the first two weeks of feed resupply (weeks 3–4) were significantly higher than the control group (F = 9.731, df = 3, *p* < 0.005; Figure 4). FEs of F5, F10, and F14 during the second two weeks (weeks 5–6) and third two weeks (weeks 7–8) of feed resupply were not significantly different from the control (F = 1.075, df = 3, *p* = 0.413; and, F = 0.812, df = 3, *p* = 0.522, respectively). The overall FE during the six weeks of feed resupply (weeks 3–8) was only significantly higher for F14 compared to the control group (F = 8.139, df = 3, *p* = 0.008).

The condition factor (CF), coefficient of variation for body length (CVBL), and body weight (CVBD) of the four experimental feeding groups at the beginning and end of the experiment are shown in Table 1. The fasting period did not affect the CF_f_ (F = 0.06, df = 3, *p* = 0.98), CVBL_f_ (F = 0.143, df = 3, *p* = 0.931) and CVBW_f_ (F = 0.086, df = 3, *p* = 0.966).

The body compositions, HSI, and VSI of *S. scherzeri* after fasting (end of week 2) and at the end of the experiment (end of week 8) are shown in Table 2. Lipid (F = 37.407, df = 3, *p* = 0.002), HSI (F = 7.612, df = 3, *p* = 0.04), of F10 and F14 after fasting (week 2), as well as lipid/LBM (F = 36.028, df = 3, *p* = 0.002) and VSI (F = 14.549, df = 3, *p* = 0.013) of all fasting groups, were significantly lower than the control; however, moisture content (F = 12.493, df = 3, *p* = 0.017) in F14 was significantly higher than the control. No differences were detected between protein (F = 2.351, df = 3, *p* = 0.214) and ash (F = 0.282, df = 3, *p* = 0.837) among experimental groups in week 2. Similarly, no significant differences were observed in moisture content (F = 8.918, df = 3, *p* = 0.06), protein (F = 0.10, df = 3, *p* = 0.96), lipid (F = 0.265, df = 3, *p* = 0.849), ash (F = 3.417, df = 3, *p* = 0.073), lipid/LBM (F = 0.117, df = 3, *p* = 0.948), and VSI (F = 0.463, df = 3, *p* = 0.716) compared to the control group at the end of experimentation; however, HSI values in F5, F10, and F14 were significantly higher than the control group (F = 0.059, df = 3, *p* = 0.016).

The Hb, GLU, TCHO, GOT, GPT, and HDLC concentrations in the blood of juvenile *S. scherzeri* after fasting and at the end of experimentation are shown in Table 3. The concentrations of TCHO (F = 7.727, df = 3, *p* = 0.01) in F14 were significantly lower than that in the control group after fasting (end of week 2), whereas concentrations of GLU in F14 were significantly higher than the control (F = 7.393, df = 3, *p* = 0.001). The concentrations of Hb (F = 0.379, df = 3, *p* = 0.771), TCHO (F = 0.239, df = 3, *p* = 0.868), GOT (F = 2.132, df = 3, *p* = 0.149), GPT (F = 2.851, df = 3, *p* = 0.082), GLU (F = 0.028, df = 3, *p* = 0.993), and HDLC (F = 0.463, df = 3, *p* = 0.988) in the blood of juvenile *S. scherzeri* at the end of experimentation were not significantly different from the control group.

## 4. Discussion

In modern aquaculture, producers are seeking to reduce production costs in a manner without adversely affecting productivity or the environment [4]. As feed represents the largest share of aquaculture production costs [35], commercial aquaculture farms aim to develop appropriate feed supply management and saving methods [36]. Compensation growth can be used as a feed supply strategy, whereby satiation feeding after feed deprivation can lead to faster growth and greater feed efficiency, translating into improved economic efficiency by reducing feed and labor costs [37,38,39]. Depending on the strength of fish body weight recovery following fasting, the degree of compensatory growth is classified as over, complete (i.e., full), partial, and non-compensatory [5]. Notably, the degree of compensatory growth is sensitive to both the degree and duration of fasting prior to satiation feeding [5,34,40]. Here, all juvenile *S. scherzeri* subjected to five- (F5), 10- (F10), and 14-day (F14) fasting showed full compensatory growth that caught up with the weight of the control group four weeks after feed resupply. Similarly, full compensatory growth after ≤2 weeks of fasting was reported to be observed in rock bream (*Oplegnathus fasciatus*) [5], Chinese longsnout catfish (*Leiocassis longirostris*) [39], gibel carp (*Carassius auratus gibelio*) [41], and gilthead seabream (*Sparus auratus*) [42]. In addition, full compensatory growth was reported in black rockfish (*Sebastes schlegeli*) after five days of fasting [8], barramundi (*Lates calcarifer*) [43] and hybrid tilapia (*Oreochromis mossambicus* × *O. niloticus*) [44] after one week of fasting, and red seabream (*Pagrus major*) after 1–3 weeks of fasting [3]. The relative ratio of the body weight of fish subjected to fasting as compared to those fish in the control group has been previously discussed as a critical indicator of full compensatory growth [3,8,41,43]. Tian and Qin [43] determined that after fasting, full compensatory growth occurred for ≥60% of fish body weight in the control group, and the same phenomenon was reported to be observed in gibel carp [41], hybrid tilapia [44], black rockfish [8], Chinese longsnout catfish [40], gilthead seabream [42], rainbow trout (*Oncorhynchus mykiss*) [45], and rock bream [5]. During the present experiment, F5, F10, and F14 all showed full compensatory growth after fasting for 5 to 14 days for individuals with body weights of ≥~60% (61.7–81.1%) than the control group.

The SGR and FR values after fasting increased during the early feed resupply stages (weeks 3–4) with an increasing duration of food deprivation, whereas similar values to the control group were observed during the latter food resupply (weeks 7–8), comparable to results previously reported for rainbow trout [45] and rock beam [5]. Generally, these phenomena occur during the initial resupply of feed immediately following the period of deprivation due to a hyperphagic response [5,8,44], inducing compensatory growth that decreases with time [5,42] as well as a decrease in the digestibility of excess feed intake [26,32,46,47]. Therefore, during the initial resupply of food in the present study, the FE of the fasting group was very high; however, with time, the FE decreased or was not significantly different from the control group. Similarly, improved FE during feed resupply was documented for European minnow (*Phoxinus phoxinus*) [48], gibel carp [49], Chinese longsnout catfish [40], Atlantic halibut *(Hippoglossus hippoglossus*) [50], blackhead seabream (*Acanthopagrus schlegelii*) [14], and rock bream [5]. Throughout the present study, hyperphagia and improved FE tended to decrease with time, both of which were attributed to the two factors acting as major determinants of compensatory growth of *S. scherzeri*, as this tendency was seen in all fasting groups, and similar results have been observed in other previous studies as well [3,5,34]. It has been reported, however, that compensatory growth occurs primarily due to the increase in FR and not from improvements in FE during feed resupply after fasting [8,41,45,51,52,53], suggesting that the major factors driving compensatory growth differ.

In the experiment here, the CF, CVBW, and CVBL of juvenile *S. scherzeri* were not affected by compensatory growth. Similarly, Silva et al. [17] reported no differences in CF, final weight, and total length between the control group and the replicates of fasting groups. Fasting groups in this study experienced a rise in moisture content after feed deprivation, and similar to previous studies, lipid content and HSI decreased [42,43,54,55]. Using a lipostatic model, one can predict the timing of compensatory growth based on the principle that such growth occurs when the lipid/LBM decreases and ends when the lipid/LBM returns to control group levels [56]. In most animals, the lipid stores indicate the easily activated energy reserves, and the lean body mass (LBM) includes the structural tissues, so the lipid/LBM may be considered an indicator of nutritional state [56]. The lipostatic model hypothesized that restricted feeding induces changes in adiposity through compensatory feeding as a result of negative feedback due to changes in energy balance [56]. Therefore, fat loss after restricted feeding reduces the negative feedback on feed intake and elevated feed intake until normal fat levels and lipid/LBM are maintained. In other words, the lipostatic model represents that the size of body lipid reserves affects the feed intake (i.e., hyperphagic response) of fish via the negative feedback signal to the central nervous system, thereby regulating the feed intake behavior, body weight, and body composition during compensatory growth. Therefore, if the body lipid reserves of fish are repleted very slowly, the compensatory growth through the hyperphagic response can be sustained because the imbalance of the lipid/LBM is maintained for a long time [56]. It is very important to extend the period of imbalance of the lipid/LBM by controlling the repletion of body lipid storage during refeeding after restricted feeding or fasting, and further study is needed in the future. This model has been applied to Atlantic salmon (*Salmo salar*) [57] and Nile tilapia (*Oreochromis niloticus*) [58]. The present experiment found that the lipid/LBM of the F5, F10, and F14 groups were significantly lower than that of the control group following fasting (end of week 2), and no differences in SGR and lipid/LBM were observed between the fasting and control groups at the end of the experiment (end of week 8), thus indicating the efficacy of the lipostatic model. Despite this alignment here, numerous previous studies’ results were inconsistent with the lipostatic model [3,5,8,40,41,42,45,59], suggesting that such results may vary by species. Based on the results of the experiment here, the HSI after fasting significantly decreased compared to the control group as the duration of fasting increased. Silva et al. [17] reared pacamã (*Lophiosilurus alexandri*) for a total of 45 days with a 6:1 and 5:2 feed to fasting regime, and their results showed that the HSI was the lowest in the 5:2 group, which is similar the results found in the present study. As an indicator of fish nutritional status [60], the HSI is a ratio of liver to body weight and can be used to quantify the energy stored in the liver as glycogen [61]. Metabolic and regulatory processes associated with nutrients and lipogenesis originate in the liver of fish [62]. As demonstrated in the present study, the lower HSI values with longer fasting periods indicated the need for greater hepatic energy storage, and Morshedi et al. [15] reported that fasting decreased the amount of hepatic energy available, directly affecting liver weight. At the end of the experiment, the higher HSI in the fasting groups compared to the control was due to an excess accumulation of energy during the recovery process associated with feed resupply. Fish can reduce their energy requirements by reducing the mass of gastrointestinal tissues during a fasting period [63], which is consistent with the findings of the VSI reduction observed after fasting in the present study. According to Jobling [64], fasting increases the digestive tract capacity of fish, allowing them to consume more due to the hyperphagia that may occur throughout the recovery period. This would partially explain the increase in VSI of the fasting group seen at the end of the present experiment and correlate with the hyperphagic behavior observed here as well.

A hematological and biochemical assessment of fish has been used to determine their physiological status [15], and it was revealed that these factors could be affected by biotic and abiotic factors, such as age, sex, water temperature, seasonal patterns, and food intake [65]. At the end of the present study, it was found that fasting and resupplying had no notable effects on the hematological characteristics of juvenile *S. scherzeri*, and similar results were found with the Siberian sturgeon (*Acipenser baerii*) [15], Nile tilapia [66], olive flounder (*Paralichthys olivaceus*) [67], European eel (*Anguilla anguilla*) [68], red porgy (*Pagrus pagrus*) [69], and beluga (*Huso huso*) [70]. During the present study, it was observed that fasting influenced GLU and TCHO levels in the plasma of juvenile *S.*
*scherzeri*; however, no such increase in plasma GLU was observed in sea bass (*Dicentraachus labrax*) [71] or pirapitinga (*Piaractus brachypomus*) [72]. Cholesterol is a structural lipid utilized or synthesized in response to feed deprivation; however, it is also maintained during feed deprivation without increasing, decreasing, or changing depending on the species and fasting duration [72]. For the present study, the cholesterol concentrations decreased after fasting (week 2), similar to the results observed for rainbow trout [73], brown trout (*Salmo trutta*) [74] and sturgeon (*Acipenser naccarii*) [75]. Additionally, a decrease in feed intake of *S. scherzeri* due to either a decrease in feeding frequency or feeding ratio was also associated with a decrease in plasma TCHO concentrations [26,31,32]. In contrast, no change in TCHO concentrations was documented for pirapitinga [72], tamaqui (*Colossoma macropomum*) [6], and pacamã [17], whereas an increase was observed with pacu (*Piaractus mesopotamicus*) [76] and climbing perch (*Anabas testudineus*) [77], similarly suggesting variable results according to species and fasting regime. Therefore, further complementary research is necessary for the future.

## 5. Conclusions

The present experiment determined that all juvenile *S. scherzeri* (initial mean weight, 14.7 g) subjected to 5–14 days of food deprivation experienced full compensatory growth after four weeks of food resupply under our experimental conditions. Responding well to the feeding strategy of fasting and resupplying, *S. scherzeri* maintained similar body compositions, blood, morphological, and biochemical characteristics compared to the continuously fed control group within 6 weeks after resupply commenced. This information may be helpful to fish farmers who encounter adverse conditions during rearing or require practical feeding strategies to increase feed efficiency while reducing costs.

## Figures and Tables

**Figure 1 animals-12-00889-f001:**
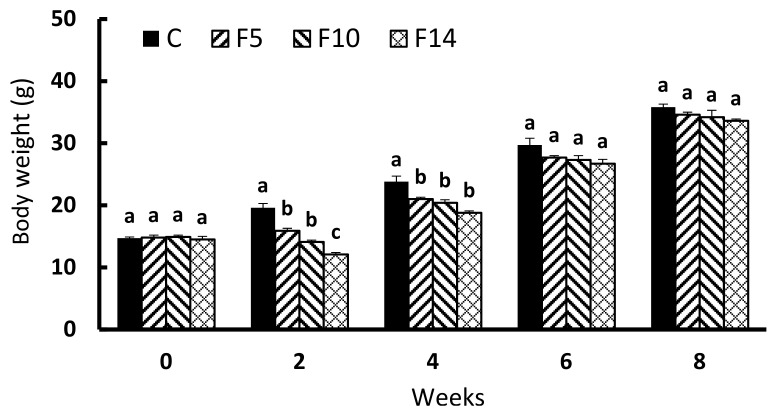
Changes in body weight of juvenile leopard mandarin fish (*Siniperca scherzeri*) subjected to four feeding regimens across an 8-week trial: Control (C), fish fed continuously; F5, fish fasted for 5 days (days 10–14); F10, fish fasted for 10 days (days 5–14); and, F14, fish fasted for 14 days (days 1–14), prior to all fasted fish being fed to satiation. Values (mean ± standard error, *n* = 5 in weeks 0 and 2, *n* = 3 in weeks 4, 6 and 8) in the same column with dissimilar letters are significantly different (*p* < 0.05).

**Figure 2 animals-12-00889-f002:**
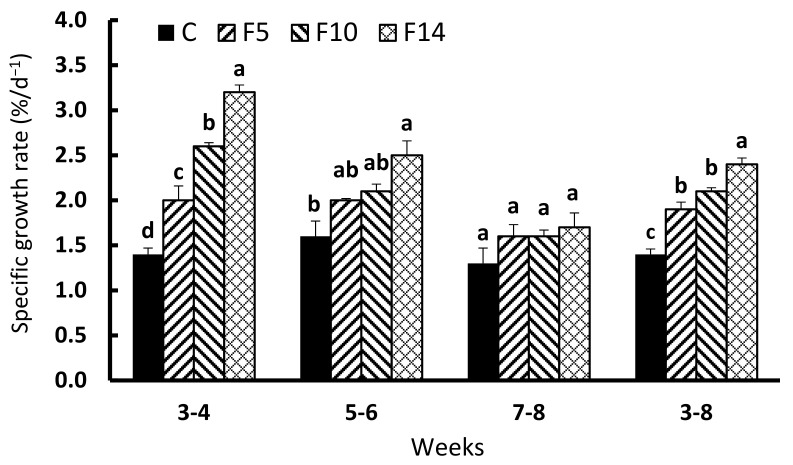
Changes in specific growth rate of juvenile leopard mandarin fish (*Siniperca scherzeri*) subjected to four feeding regimens across an 8-week trial: Control (C), fish fed continuously; F5, fish fasted for 5 days (days 10–14); F10, fish fasted for 10 days (days 5–14); and, F14, fish fasted for 14 days (days 1–14), prior to all fasted fish being fed to satiation. Values (mean ± standard error, *n* = 3) in the same column with dissimilar letters are significantly different (*p* < 0.05).

**Figure 3 animals-12-00889-f003:**
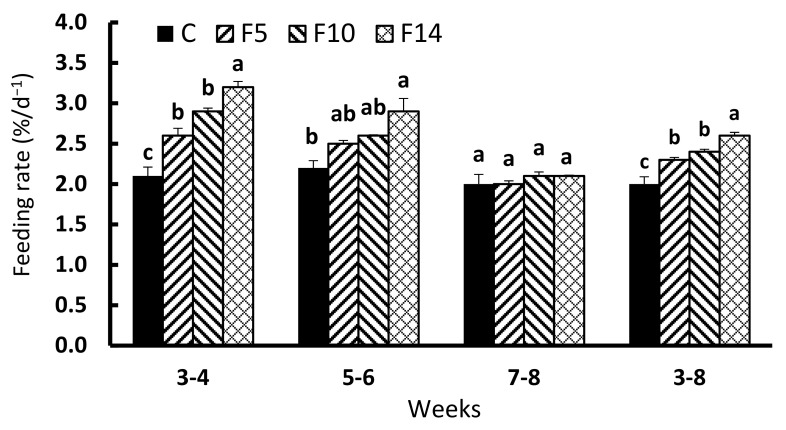
Changes in feeding rate of juvenile leopard mandarin fish (*Siniperca scherzeri*) subjected to four feeding regimens across an 8-week trial: Control (C), fish fed continuously; F5, fish fasted for 5 days (days 10–14); F10, fish fasted for 10 days (days 5–14); and, F14, fish fasted for 14 days (days 1–14), prior to all fasted fish being fed to satiation. Values (mean ± standard error, *n* = 3) in the same column with dissimilar letters are significantly different (*p* < 0.05).

**Figure 4 animals-12-00889-f004:**
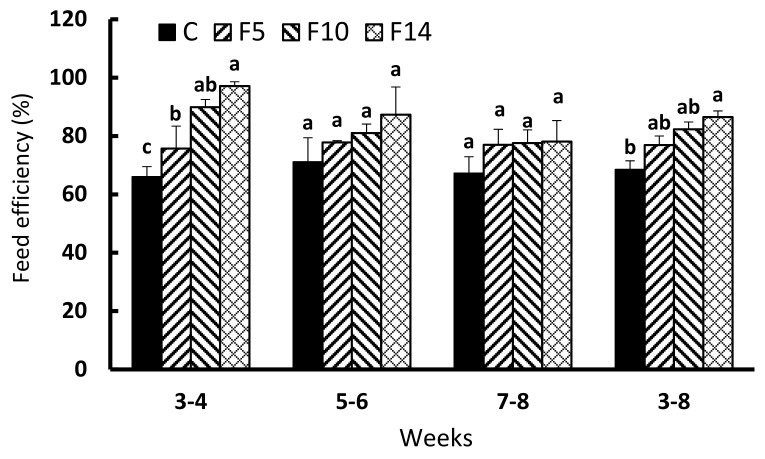
Changes in feed efficiency of juvenile leopard mandarin fish (*Siniperca scherzeri*) subjected to four feeding regimens across an 8-week trial: Control (C), fish fed continuously; F5, fish fasted for 5 days (days 10–14); F10, fish fasted for 10 days (days 5–14); and, F14, fish fasted for 14 days (days 1–14), prior to all fasted fish being fed to satiation. Values (mean ± standard error, *n* = 3) in the same column with dissimilar letters are significantly different (*p* < 0.05).

**Table 1 animals-12-00889-t001:** Condition factor (CF), coefficient of variation for body length (CVBL), and body weight (CVBW) of juvenile leopard mandarin fish (*Siniperca scherzeri*) subjected to four different feeding regimens for 8 weeks *.

Treatment	CF_i_ (%) ^1^	CF_f_ (%) ^2^	CVBL_i_ (%) ^3^	CVBL_f_ (%) ^4^	CVBW_i_ (%) ^5^	CVBW_f_ (%) ^6^
C	1.07 ± 0.07	1.07 ± 0.03	13.6 ± 0.20	9.0 ± 0.4	18.7 ± 0.86	27.7 ± 1.13
F5	1.05 ± 0.01	1.09 ± 0.03	13.4 ± 0.19	8.1 ± 1.1	13.7 ± 1.24	27.2 ± 1.02
F10	1.06 ± 0.02	1.07 ± 0.03	13.7 ± 0.16	7.8 ± 0.9	15.3 ± 1.81	27.2 ± 0.81
F14	1.07 ± 0.01	1.10 ± 0.03	13.6 ± 0.58	5.6 ± 0.7	16.0 ± 0.76	27.5 ± 0.52

* Values (mean ± standard error, *n* = 5 in CF_i_, CVBL_i_ and CVBW_i_, *n* = 3 in CF_f_, CVBL_f_ and CVBW_f_) are not significantly different for any treatment (*p* > 0.05). ^1^ CF_i_ (%) = (fish initial weight/(fish initial length)^3^) × 100. ^2^ CF_f_ (%) = (fish final weight of fish/(fish final length)^3^) × 100. ^3^ CVBL_i_ (%) = (standard deviation of initial fish length/mean initial fish length) × 100. ^4^ CVBL_f_ (%) = (standard deviation of final fish length/mean final fish length) × 100. ^5^ CVBW_i_ (%) = (standard deviation of initial fish weight/mean initial fish weight) × 100. ^6^ CVBW_f_ (%) = (standard deviation of final fish weight/mean final fish weight of fish) × 100.

**Table 2 animals-12-00889-t002:** Whole-body proximate composition (%, wet weight basis), hepatosomatic index (HSI), and viscerasomatic index (VSI) of juvenile leopard mandarin fish (*Siniperca scherzeri*) at the end of week 2 and week 8 in four different feeding groups *.

Periods	Variables	Treatment			
		C	F5	F10	F14
Week 2 (*n* = 2)	Moisture (%)	70.6 ± 0.9 ^b^	73.5 ± 0.6 ^ab^	74.8 ± 0.2 ^ab^	77.3 ± 1.2 ^a^
	Crude protein (%)	19.9 ± 0.8	18.3 ± 0.3	20.0 ± 0.2	19.9 ± 0.9
	Crude lipid (%)	4.7 ± 0.05 ^a^	3.7 ± 0.05 ^b^	3.5 ± 0.05 ^b^	2.9 ± 0.15 ^c^
	Ash (%)	6.1 ± 0.1	6.0 ± 0.1	6.1 ± 0.1	6.1 ± 0.1
	HSI (%) ^1^	2.5 ± 0.08 ^a^	1.7 ± 0.37 ^ab^	1.4 ± 0.14 ^b^	1.3 ± 0.06 ^b^
	VSI (%) ^2^	7.7 ± 0.40 ^a^	5.0 ± 0.22 ^b^	3.9 ± 0.55 ^c^	3.5 ± 0.14 ^c^
	Lipid/LBM ^3^	0.18 ± 0.01 ^a^	0.15 ± 0.01 ^b^	0.13 ± 0.01 ^bc^	0.10 ± 0.01 ^c^
Week 8 (*n* = 3)	Moisture (%)	69.5 ± 0.4	71.8 ± 0.4	72.2 ± 0.4	72.0 ± 0.2
	Crude protein (%)	19.5 ± 0.7	19.5 ± 0.3	19.8 ± 0.2	19.7 ± 0.5
	Crude lipid (%)	3.7 ± 0.2	3.6 ± 0.4	3.4 ± 0.3	3.6 ± 0.3
	Ash (%)	6.5 ± 0.2	6.0 ± 0.3	5.6 ± 0.2	5.9 ± 0.1
	HSI (%) ^1^	1.3 ± 0.06 ^b^	1.6 ± 0.08 ^a^	1.7 ± 0.09 ^a^	1.7 ± 0.02 ^a^
	VSI (%) ^2^	4.6 ± 0.06	4.5 ± 0.28	4.6 ± 0.19	4.8 ± 0.21
	Lipid/LBM ^3^	0.14 ± 0.01	0.14 ± 0.02	0.13 ± 0.01	0.14 ± 0.01

* Values (mean ± standard error) with different superscripts in the same row are significantly different (*p* < 0.05). ^1^ HSI (%) = (liver weight/fish weight) × 100. ^2^ VSI (%) = (liver weight/fish weight) × 100. ^3^ Lipid/LBM, ratio of lipids to sum of protein and ash.

**Table 3 animals-12-00889-t003:** Plasma chemical composition of juvenile leopard mandarin fish (*Siniperca scherzeri*) at the end of week 2 and week 8 week in four different feeding groups *.

Periods		C	F5	F10	F14
Week 2 (*n* = 2)	Hb (g dL^−1^) ^1^	7.8 ± 0.28	8.0 ± 0.36	7.7 ± 0.17	7.8 ± 0.15
	GLU (mg dL^−1^) ^2^	150.3 ± 12.7 ^b^	141.0 ± 14.2 ^b^	177.3 ± 14.3 ^ab^	232.9 ± 17.2 ^a^
	TCHO (mg dL^−1^) ^3^	170.4 ± 3.55 ^a^	169.7 ± 4.07 ^a^	159.9 ± 3.25 ^ab^	150.0 ± 2.82 ^b^
	GOT (U L^−1^) ^4^	395.1 ± 23.7	387.1 ± 14.2	375.5 ± 18.2	333.3 ± 10.6
	GPT (U L^−1^) ^5^	129.2 ± 5.2	143.9 ± 6.6	150.1 ± 7.0	139.6 ± 6.0
	HDLC (U L^−1^) ^6^	108.2 ± 1.60	109.1 ± 0.77	107.9 ± 1.51	108.1 ± 0.94
Week 8 (*n* = 3)	Hb (g dL^−1^) ^1^	7.3 ± 0.9	7.5 ± 0.6	6.6 ± 0.6	6.7 ± 0.5
	GLU (mg dL^−1^) ^2^	138.0 ± 10.1	141.0 ± 8.5	139.7 ± 6.36	138.3 ± 7.33
	TCHO (mg dL^−1^) ^3^	143.3 ± 8.5	143.8 ± 3.3	137.0 ± 1.7	143.0 ± 8.3
	GOT (U L^−1^) ^4^	226.8 ± 14.0	249.0 ± 15.1	274.5 ± 11.6	223.3 ± 9.5
	GPT (U L^−1^) ^5^	71.0 ± 8.6	64.6 ± 5.4	53.5 ± 2.5	44.3 ± 2.7
	HDLC (U L^−1^) ^6^	98.5 ± 5.1	99.8 ± 2.7	98.0 ± 0.9	98.3 ± 4.5

* Values (mean ± standard error) with different superscripts in the same row are significantly different (*p* < 0.05). ^1^ Hb = hemoglobin. ^2^ GLU = glucose. ^3^ TCHO = total cholesterol. ^4^ GOT = glutamic oxaloacetic transaminase. ^5^ GPT = glutamic pyruvic transaminase. ^6^ HDLC = high density lipoprotein cholesterol.

## Data Availability

The datasets generated and/or analyzed during the current study are available from the corresponding author on reasonable request.

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
