# Peer review of "Effect of Fasting and Refeeding on Juvenile Leopard Mandarin Fish Siniperca scherzeri"

_animals, 2022, doi:10.3390/ani12070889_

Round 1
Reviewer 1 Report
Dear authors, below are my considerations about the manuscript:
Title: I suggest a shorter title that is more attractive to the reader.
Keywords: I suggest you use different words than those used in the title.
Introduction:
- Line 58: What commercial weight does this species reach in captivity? Please enter more information about this in the introduction.
- Line 64: What is the difference between feeding ratio and feed rate? Please clarify.
Material and methods:
Overall this section is very confusing.
- Lines 81-83: Include a citation that shows these are ideal water quality conditions for the species.
- Lines 85-94: How long was the experiment? 8 weeks? This paragraph is very confusing.
- Lines 105-125: State in detail how the fish were euthanized. If possible, provide your institution's ethics committee approval number.
Results:
- Tables: I suggest you follow the same reasoning for the letters as used in the figures. Letter "a", higher value and letter "c", lower value.
Discussion:
Overall, authors need to further discuss hematologic parameters, especially these large differences in glucose levels (see table in results). How to explain this difference in glucose levels between weeks 2 and 8 in the control group?
An analysis of blood triglycerides could help the authors to better explain the decrease in HSI (possible lipid mobilization in the liver) related to the results obtained for body lipids. Body lipids clearly show that this species used its lipid reserves as a source of energy during food deprivation.
- Line 323: Please provide more details about this lipostatic model and its importance in restriction studies.
Conclusion:
- With these results, would the authors recommend that the producer use a 14-day fasting protocol? Do you really believe that it would be viable to produce this species?
Author Response
Title: I suggest a shorter title that is more attractive to the reader.
Answer: Following the reviewer’s suggestion, we have revised the title to “Effect of fasting and refeeding on juvenile leopard mandarin fish Siniperca scherzeri”.
Keywords: I suggest you use different words than those used in the title.
Answer: Following the reviewer’s suggestion, we have replaced with new Keywords as follows: aquaculture, body composition, blood chemistry, starvation, compensatory growth
Line 58: What commercial weight does this species reach in captivity? Please enter more information about this in the introduction.
Answer: This species is usually harvested from fish farms after growing from 100 g to over 300 g depending on the consumer's preference (i.e., for soup or sashimi).
Therefore, we have inserted following sentence: This species is harvested from fish farms in commercial size ranging from relatively small 100 g to over 300 g depending on consumer preferences in Korea.
Line 64: What is the difference between feeding ratio and feed rate? Please clarify.
Answer: Following the reviewer’s suggestion, we have revised “feeding ratio” to “feeding ratio for satiation level” and “feeding rate” to “feeding rate for fish weight”.
Lines 81-83: Include a citation that shows these are ideal water quality conditions for the species.
Answer: Although there have not been any studies on the effect of water quality on this species, it was comparable to the range of water quality shown in previous studies with same species (Kim et al., 2021[26], Kim et al., 2020[31], Kim et al., 2021[32]). In addition, it was found that most of the water quality standards for aquaculture suggested by Meade (1989) were met, so it is considered that there is no effect by water quality during the period of this study. As suggested by the reviewer, studies on ideal water quality conditions are needed in the future.
Meade JW. 1989. Aquaculture Management. New York: Van Nostrand Reinhold.
Lines 85-94: How long was the experiment? 8 weeks? This paragraph is very confusing.
Answer: Following the reviewer’s comment, Materials and Methods was substantially changed and included adequate information. In addition, the total duration of the experiment was also specified as 8 weeks.
Lines 105-125: State in detail how the fish were euthanized. If possible, provide your institution's ethics committee approval number.
Answer: Following the reviewer’s comment, we have inserted the detailed explanation about euthanizing fish (Line 120-122), “Fish were sacrificed by anesthetizing in 150 mg/L 2-phenoxyethanol solution for 1 minute after measurement of body weight or body length, and stored at -40ºC for the analysis of body composition.”
”.
The institution’s ethics committee approval number is already presented in Institutional Review Board Statement as follows; This research was approved by Inha University Institutional Animal Use and Care Committee (INHA 210811-785).
Tables: I suggest you follow the same reasoning for the letters as used in the figures. Letter "a", higher value and letter "c", lower value.
Answer: Following the reviewer’s comment, we have revised Letter “a, b and etc” in Table as well as in Figure.
Overall, authors need to further discuss hematologic parameters, especially these large differences in glucose levels (see table in results). How to explain this difference in glucose levels between weeks 2 and 8 in the control group?
Answer: We apologize for the misrepresentation of the glucose value in control at week 8 as well as those of other feeding groups (i.e. F5, F10 and F14) at week 8 in Table 3. We inserted the exact glucose value of each feeding groups at week 8, and the value of the control in week 8 did not show a significant difference from that of control in week 2. We apologize again for our mistakes.
An analysis of blood triglycerides could help the authors to better explain the decrease in HSI (possible lipid mobilization in the liver) related to the results obtained for body lipids. Body lipids clearly show that this species used its lipid reserves as a source of energy during food deprivation.
Answer: We fully agree with the reviewer’s opinion, but unfortunately we did not measure blood triglyceride. Triglyceride values can help in the adequate interpretation of changes in body lipid content and HSI value after fasting and refeeding, but we don’t have triglyceride data in this study. Therefore, it is considered inappropriate to mention it in the Discussion. However, we believe that an appropriate explanation can be obtained from the results of body lipid and HSI presented in this study.
Line 323: Please provide more details about this lipostatic model and its importance in restriction studies.
Answer: Following the reviewer’s comment, we have inserted the detailed explanation as follows: In most animals, the lipid stores indicate the easily activated energy reserves, and the lean body mass (LBM) include the structural tissues, so the lipid/LBM may consider an indicator of nutritional state. The lipostatic model hypothesized that restricted feeding induces changes in adiposity through compensatory feeding as a result of negative feedback due to changes in energy balance. Therefore, fat loss after restricted feeding reduces negative feedback on feed intake and elevated feed intake until normal fat levels and lipid/LBM are maintained. In other word, the lipostatic model represents that the size of body lipid reserves affects the feed intake (i.e. hyperphagic response) of fish via to the negative feedback signal to the central nervous system, thereby regulates the feed intake behavior, body weight and body composition during compensatory growth. Therefore, if the body lipid reserves of fish are repleted very slowly, the compensatory growth through hyperphagic response can be sustained because the imbalance of the lipid/LBM is maintained for a long time. It is very important to extend the period of imbalance of the lipid/LBM by controlling the repletion of body lipid storage during refeeding after restricted feeding or fasting, and further study is needed in the future.
With these results, would the authors recommend that the producer use a 14-day fasting protocol? Do you really believe that it would be viable to produce this species? Answer: We carried out experiments under limited conditions such as fish size (about 14-36 g), water temperature (26.4-27.5℃), feed (55.4% protein and 14.1% lipid) and fasting period (5-14 days). Therefore, we have given as “under our experimental conditions” in both Abstract and Conclusion to prevent extrapolate the findings beyond experimental conditions.
During commercial aquaculture practices, fish farmers intentionally or non-intentionally choose fasting or limited feeding for prolonged periods to save feed, enhance the growth rate, and decrease mortality caused by environmental deterioration, diseases and handling for selection or transport. Therefore, we believe that fasting and refeeding for appropriate periods could induce a significant improvement of feed efficiency and specific growth rate of juvenile mandarin fish based on our results, and it could be applicable as a feeding strategy in commercial farming.

Reviewer 2 Report
The Authors undertook the issue of compensatory growth after various starvation periods and subsequent refeeding in Siniperca scherzeri. The Introduction provides clear information about the significance of the studied fish species and clearly stated purpose of the study. M&M lack an explicit description how fish feeding was evaluated - if they were fed to satiation - how the feeding was performed, how satiation was observed and how the amount of consumed food was measured. In description of statistical analysis it is mentioned that n=5 or n=3 were used, while in the previous chapter (Analysis of blood and body content) the sentence "A total of two rearing tanks were randomly selected from each of the five replicates of the four feeding groups, and seven fish per tank were subjected to blood composition, HSI, and VSI analyses, while eight fish were subjected to the whole- body analysis" which suggests that for blood analysis n=14 and for HSI and VSI n=16. Please, add appropriate n numbers in each figure and table description. In Table 3 very high differences in glucose concentrations (GLU) between 2 and 8 week in all fish groups are shown (e.g. 150.3 and 38.0 for the control). How the Authors explain such a decrease? Please, add an appropriate paragraph in the Discussion.
Author Response
The Authors undertook the issue of compensatory growth after various starvation periods and subsequent refeeding in Siniperca scherzeri. The Introduction provides clear information about the significance of the studied fish species and clearly stated purpose of the study.
M&M lack an explicit description how fish feeding was evaluated - if they were fed to satiation - how the feeding was performed, how satiation was observed and how the amount of consumed food was measured.
Answer: Following the reviewer’s comment, we have inserted the detailed explanation about satiation level, feeding and the amount of consumed food as follows: Daily feeding was performed by dropping the pellet (the same as during the acclimation period) into each tank every few minutes until the fish ceased to eat due to satiation. Care was taken to ensure that all pellets provided was consumed by the fish. The container of the pellets was weighed before the first feeding and after the last feeding daily. Uneaten pellets were collected from each tank by siphoning and dried in oven to a constant weight. Actual daily consumed feed amount was determined as the difference in weight between the feed supplied into the tank and the uneaten feed removed by siphon.
In description of statistical analysis it is mentioned that n=5 or n=3 were used, while in the previous chapter (Analysis of blood and body content) the sentence "A total of two rearing tanks were randomly selected from each of the five replicates of the four feeding groups, and seven fish per tank were subjected to blood composition, HSI, and VSI analyses, while eight fish were subjected to the whole- body analysis" which suggests that for blood analysis n=14 and for HSI and VSI n=16. Please, add appropriate n numbers in each figure and table description.
Answer: Following the reviewer’s comment, we have inserted as follows in Materials and Methods: Each replicate was considered as an experimental unit, and therefore, the mean value obtained from a replicate within each feeding group was used as a data unit.
According to mentioned above, the mean value of blood, HSI and VSI samples within each replicate was used to as data unit. We inserted appropriate n numbers in each figure and table description
In Table 3 very high differences in glucose concentrations (GLU) between 2 and 8 week in all fish groups are shown (e.g. 150.3 and 38.0 for the control). How the Authors explain such a decrease? Please, add an appropriate paragraph in the Discussion.
Answer: We apologize for the misrepresentation of the glucose value in control at week 8 as well as those of other feeding groups (i.e. F5, F10 and F14) at week 8 in Table 3. We inserted the exact glucose value of each feeding groups at week 8, and the value of the control in week 8 did not show a significant difference from that of control in week 2. We apologize again for our mistakes.

Round 2
Reviewer 1 Report
Thanks for the replies. I accept in the present form.